# Quercetin Ameliorates Deoxynivalenol-Induced Intestinal Injury and Barrier Dysfunction Associated with Inhibiting Necroptosis Signaling Pathway in Weaned Pigs

**DOI:** 10.3390/ijms242015172

**Published:** 2023-10-14

**Authors:** Jiahao Liu, Mohan Zhou, Qilong Xu, Qingqing Lv, Junjie Guo, Xu Qin, Xiaoye Xu, Shaokui Chen, Jiangchao Zhao, Kan Xiao, Yulan Liu

**Affiliations:** 1Hubei Key Laboratory of Animal Nutrition and Feed Science, Wuhan Polytechnic University, Wuhan 430023, China19356055812@163.com (X.Q.); 15549416907@163.com (X.X.); loveskchen@163.com (S.C.); 2Department of Animal Science, Division of Agriculture, University of Arkansas, Fayetteville, AR 72701, USA; jzhao77@uark.edu

**Keywords:** Quercetin, deoxynivalenol, intestinal injury, barrier dysfunction, necroptosis signaling pathway

## Abstract

Quercetin (Que) is a flavonol compound found in plants, which has a variety of biological activities. Necroptosis, a special form of programmed cell death, plays a vital role in the development of many gastrointestinal diseases. This study aimed to explore whether Que could attenuate the intestinal injury and barrier dysfunction of piglets after deoxynivalenol (DON) exposure through modulating the necroptosis signaling pathway. Firstly, twenty-four weaned piglets were used in a 2 × 2 factorial design and the main factors, including Que (basal diet or diet supplemented with 100 mg/kg Que) and DON exposure (control feed or feed contaminated with 4 mg/kg DON). After feeding for 21 d, piglets were killed for samples. Next, the intestinal porcine epithelial cell line (IPEC-1) was pretreated with or without Que (10 μmol/mL) in the presence or absence of a DON challenge (0.5 μg/mL). Dietary Que increased the body weight, average daily gain, and average daily feed intake (*p* < 0.05) through the trial. Que supplementation improved the villus height, and enhanced the intestinal barrier function (*p* < 0.05) indicated by the higher protein expression of occludin and claudin-1 (*p* < 0.05) in the jejunum of the weaned piglets after DON exposure. Dietary Que also down-regulated the protein abundance of total receptor interacting protein kinase 1 (t-RIP1), phosphorylated RIP1 (p-RIP1), p-RIP3, total mixed lineage kinase domain-like protein (t-MLKL), and p-MLKL (*p* < 0.05) in piglets after DON exposure. Moreover, Que pretreatment increased the cell viability and decreased the lactate dehydrogenase (LDH) activity (*p* < 0.05) in the supernatant of IPEC-1 cells after DON challenge. Que treatment also improved the epithelial barrier function indicated by a higher transepithelial electrical resistance (TEER) (*p* < 0.001), lower fluorescein isothiocyanate-labeled dextran (FD4) flux (*p* < 0.001), and better distribution of occludin and claudin-1 (*p* < 0.05) after DON challenge. Additionally, pretreatment with Que also inhibited the protein abundance of t-RIP1, p-RIP1, t-RIP3, p-RIP3, t-MLKL, and p-MLKL (*p* < 0.05) in IPEC-1 cells after DON challenge. In general, our data suggest that Que can ameliorate DON-induced intestinal injury and barrier dysfunction associated with suppressing the necroptosis signaling pathway.

## 1. Introduction

Mycotoxins are the most common natural contaminants existing in human food and animal feed. Among the mycotoxins, deoxynivalenol (DON), predominantly generated by *Fusarium graminearum* and *F. culmorum*, is often encountered in cereals and cereal products [1]. DON contamination has brought a great risk to human and animal health as it presents in common cereals or by-products [1]. Among domestic animals, swine exhibits the highest sensitivity to DON. The toxic effects caused by DON include impaired intestinal barrier function, immunosuppression, growth retardation, and reproductive disorders [2]. There is abundant evidence that DON damages the intestinal health and disrupts intestinal barrier function, leading to compromised growth performance and gut diseases [3]. Recently, it was found that the intestinal toxicity of DON was associated with regulating mitochondrial homeostasis or cell death; however, the molecular mechanisms are still not clear [4].

Necroptosis is a specifical form of programmed cell death, which contributes to the pathogenesis of many diseases [5,6]. Necroptosis is a tightly regulated inflammatory form of cell death, accompanied by the spread of inflammation [6]. Necroptosis morphologically exhibits the features of necrosis; however, it exhibits a unique signaling pathway that requires the involvement of receptor interaction protein kinase (RIP) 1 and RIP3, and mixed lineage kinase domain-like protein (MLKL) [6]. Currently, necroptosis has been verified as playing an important role in the gut homeostasis and inflammation caused by multiple factors [7,8]. Until now, there is little research about the effects of nutritional strategy on the necroptosis signaling pathway. 

Flavonoids are part of the polyphenol family of phytonutrients, which are mostly found in flowers, fruits, and vegetables [9]. They are commonly used in human health products because of their potent antioxidant effects [10]. Quercetin (Que, C_15_H_10_O_7_), a kind of polyhydroxy flavone, is the most common dietary flavonoid [11]. Recently, considerable attention has been paid to Que because of its potential beneficial effects, including anti-inflammatory, anti-tumor, anti-viral, anti-allergic, anti-antioxidant, and immunomodulatory effects [12]. Emerging studies have found that dietary supplementation with Que attenuated intestinal damage and inflammation in piglets during long-distance transportation [13]. However, the molecular mechanisms of the beneficial role of Que on intestinal health are still little-known. 

Accordingly, we hypothesized that Que could attenuate intestinal injury and barrier dysfunction via suppressing the necroptosis signaling pathway in weaned pigs. Pigs are the most susceptible to DON exposure among all the animals. Therefore, we firstly established an intestinal injury model by feeding piglets with DON-contaminated feed and then treated IPEC-1 cells with DON to establish an in vitro intestinal injury model. The aim of this study was to investigate the beneficial role of Que on intestinal injury and barrier function caused by DON exposure and further explore the molecular mechanisms. 

## 2. Results

### 2.1. Effects of Dietary Que on Growth Performance in Weaned Piglets after DON Exposure

Throughout the 21-d trial, piglets fed with DON had a lower final body weight (BW) (*p* < 0.05), average daily gain (ADG) (*p* < 0.05), and average daily feed intake (ADFI) (*p* < 0.05) (Figure 1A–P). Que × DON interaction was observed for the feed: gain ratio (F/G) (*p* < 0.05) in days 1–7, in which Que supplementation was associated with a trend of decreasing the F/G in piglets fed with DON; however, there was no difference in non-DON fed piglets. Piglets fed with DON also had a lower ADFI (*p* < 0.05) on days 8–14, ADG (*p* < 0.05) and ADFI (*p* < 0.05) on days 15–21. However, piglets fed Que tended to have a higher ADG (*p* = 0.051) and lower F/G (*p* = 0.087) on days 15–21. 

### 2.2. Effects of Dietary Que on Intestinal Morphology in Weaned Piglets after DON Exposure

Compared to the control group, DON exposure led to intestinal morphologic damage demonstrated as the lifting of the epithelium at the tip of the villus, villous atrophy, and hemorrhage in the lamina propria (Figure 2A). However, Que supplementation rescued these intestinal morphologic changes after DON exposure. Furthermore, DON exposure also caused intestinal ultrastructure injury including intercellular space enlargement, endoplasmic reticulum expansion, mitochondria swelling, and cristae disappearance as well as nuclear deformation and nuclear membrane rupture compared with the control group (Figure 2B); however, dietary Que attenuated the ultrastructure injury after DON exposure in the piglets. The piglets exposed to DON had a lower villus height (*p* = 0.001) than the non-DON-fed piglets (Figure 2C,D). Compared to the piglets fed the basal diet, pigs fed with Que have a higher jejunal villus height (*p* < 0.05). A Que × DON interaction was observed for the villus height (*p* < 0.1). Que supplementation increased the villus height in the piglets fed with DON, whereas there was no effect among the non-DON-fed piglets. However, neither DON nor Que affected the villus height/crypt depth.

### 2.3. Effects of Dietary Que on Intestinal Tight Junction Protein Expression in Weaned Piglets after DON Exposure

The piglets fed with DON had a lower claudin-1 and occludin (*p* < 0.05) protein expression than the non-DON-fed piglets (Figure 3). A Que × DON interaction was observed for occludin, in which Que supplementation increased the occludin protein expression in the piglets exposed with DON, whereas no difference was found for occludin among the non-DON-fed piglets. No Que × DON interaction was observed for claudin-1, in which Que supplementation increased the claudin-1 protein expression both in the DON-exposed piglets and the non-DON-fed piglets.

### 2.4. Effects of Dietary Que on Intestinal Necroptosis Protein Expression in Weaned Piglets after DON Exposure

Piglets fed with DON had a higher total receptor interacting protein 1 (t-RIP1), phosphorylated receptor interacting protein 1 (p-RIP1), total receptor interacting protein 3 (p-RIP3), total mixed-lineage kinase domain-like protein (t-MLKL), and phosphorylated mixed-lineage kinase domain-like protein (p-MLKL) protein expression (*p* < 0.05) than the non-DON-fed piglets (Figure 4). Que × DON interactions were observed for t-RIP1, p-RIP1, phosphorylated receptor interacting protein 3 (p-RIP3), t-MLKL, and p-MLKL protein expression in which Que supplementation decreased the t-RIP1, p-RIP1, p-RIP3, t-MLKL, and p-MLKL protein expression in piglets fed with DON, whereas no difference was found for the t-RIP1 and t-MLKL protein abundance among the non-DON-fed piglets. However, neither DON nor Que affected the t-RIP3 protein abundance and the ratios of p-RIP1/t-RIP1, p-RIP3/t-RIP3, and p-MLKL/t-MLKL.

### 2.5. Effects of Que on Cell Viability and Lactate Dehydrogenases (LDH) Activity in IPEC-1 Cell after DON Challenge 

The cells treated with DON had a lower cell viability and higher LDH activity in the cell supernatant (*p* < 0.001) than the non-DON-treated cells (Figure 5A,B). Que × DON interactions were observed for the cell viability and LDH activity in which Que supplementation increased the cell viability and decreased the LDH activity in the cells treated with DON, whereas no difference was found in the cell viability and the LDH activity among the non-DON-treated cells.

### 2.6. Effects of Que on Cell Barrier Integrity in IPEC-1 Cells after DON Challenge 

The cells treated with DON had a lower transepithelial electrical resistance (TEER) and higher fluorescein isothiocyanate (FITC)-labeled dextran 4 kDa (FD4) flux from the basal chamber to the apical chamber (*p* < 0.001) than the non-DON-treated cells (Figure 5C,D). A Que × DON interaction was observed for the TEER in which Que supplementation increased the TEER in IPEC-1 cells treated with DON; however, no difference was found for the TEER among the non-DON-treated cells. No Que × DON interaction was observed for the FD4 flux in which Que supplementation reduced the FD4 flux in the cells treated with DON; however, no difference was found for the FD4 flux among the non-DON-treated cells.

### 2.7. Effects of Que on Tight Junction Protein Distribution in IPEC-1 Cells after DON Challenge

The DON challenge disrupted the distribution of the tight junction proteins occludin and claudin-1 (Figure 6). The cells treated with DON had a lower claudin-1 and occludin fluorescence intensity (*p* < 0.05) than the non-DON-treated cells. Que × DON interactions were observed for the claudin-1 and occludin fluorescence intensity in which Que supplementation increased the claudin-1 and occludin fluorescence intensity in IPEC-1 cells treated with DON; however, no difference was found for the claudin-1 and occludin fluorescence intensity among the non-DON-treated cells. 

### 2.8. Effects of Que on Necroptosis Protein Expression in IPEC-1 Cells after DON Challenge

The cells treated with DON had a higher t-RIP1, p-RIP1, t-MLKL, and p-MLKL protein expression, and p-MLKL/p-MLKL ratio (*p* < 0.05) than the non-DON-treated cells (Figure 7). Que × DON interactions were observed for the p-RIP1, t-RIP1, and p-MLKL protein expression in which Que supplementation decreased p-RIP1, t-RIP1, p-MLKL, and t-MLKL protein expression in the cells treated with DON, whereas no difference was found for the p-RIP1, t-RIP1, p-MLKL, and t-MLKL protein abundance among the non-DON-treated cells. However, neither DON nor Que affected the t-RIP3 and p-RIP3 protein abundance, and the ratios of p-RIP1/t-RIP1 and p-RIP3/t-RIP3 in cells.

## 3. Discussion

DON contamination has been found in commonly consumed cereals and cereal by-products, which has posed a great threat to human and animal health [1]. The intestinal tract is the primary target for DON. After DON exposure, the intestinal barrier is the first line of defense against mycotoxin, which means that the intestine is exposed to a higher concentration of DON than other organs [14]. Increasing evidence indicates DON can lead to intestinal injury and cause barrier dysfunction [15,16]. DON is generated from *Fusarium* species; thus, the main method used to eliminate the impacts of DON was through azole insecticides to inhibit the growth of *Fusarium*. However, the disadvantages of the insecticide are evident, including residues in animals and environmental pollution [17,18]. Quercetin is a flavonoid widely found in vegetables and fruits, which has been regarded as a dietary antioxidant in food products [19,20]. Emerging evidence has shown that Que has various biological functions such as anti-inflammatory, anti-virus, anti-oxidative, and anti-bacterial activities [19,21]. This experiment aimed to explore the preventative effects of Que on DON-induced intestinal injury and barrier dysfunction so as to seek an effective method to relieve DON poisoning in production. 

In the present study, DON exposure decreased the final BW, ADG, and ADFI, and increased the F/G in piglets, which suggests that DON inhibited the growth performance. Our results are consistent with Wellington et al., who reported that feeding a 5 ppm (mg/kg) DON-contaminated diet to finishing pigs reduced the growth performance and final BW [22]. However, dietary Que tended to improve the growth performance. This is similar with Park et al., who found that supplementation with flavonoid significantly increased the ADG during days 0–21 in pigs after LPS challenge [23]. Liu et al. also demonstrated that intermediate levels (0.2–0.4 g/kg feed) of Que supplementation improved the feed efficiency in hens [24]. Therefore, our experiment demonstrated that dietary supplementation with Que could attenuate the decrease in growth performance in weaned piglets after DON exposure.

The intestinal barrier is the first line of defense against DON when DON-contaminated feed is consumed by human or animals. The intestine is exposed to higher concentration of toxin than other organs [25]. The intact intestinal morphology plays an important role in resisting the stress of external toxins [26]. In the present study, feeding DON-contaminated feed decreased the jejunal villus height, which suggests that DON caused intestinal histological damage in the weaned piglets. This is consistent with Wang et al., who found that piglets ingesting 1.28 mg/kg or 2.89 mg/kg of a DON-contaminated diet had a lower villus height in jejunum than the control pigs. As expected, dietary Que increased the jejunal villus height, which indicated that Que protected the intestinal morphology after DON exposure. Similarly, Sukhotnik et al. found that Que prevented small intestinal damage and enhanced intestine recovery during methotrexate-induced intestinal mucositis in rats [27]. To further investigate the effects of DON exposure on cell injury, we next conducted in vitro experiment. In agreement with in vivo experiment, we found that DON challenge decreased the cell viability and increased the LDH activity in IPEC-1 cells, which demonstrated DON caused intestinal epithelial cell injury. Our data are consistent with the results of Yang et al. [28]. and Deng et al. [29]. Moreover, pretreatment with Que also attenuated the decrease in cell viability and the increase in cell LDH release after DON treatment in IPEC-1 cells, which suggests that Que exerted protective effects on intestinal cell damage after DON challenge. This is supported by Jia et al., who found that Que attenuated diquat-induced cell injury by regulating GSH-related redox homeostasis in porcine enterocytes [30]. 

The intestinal barrier is composed of a layer of columnar epithelium and interepithelial tight junctions. The tight junction proteins, including occludin, claudins, and ZO, are known as the most important organizers of the tight junctions [31]. In the present study, dietary DON downregulated the protein expression of claudin-1 and occludin in the intestine of the piglets. However, Que supplementation increased the claudin-1 and occludin protein expression after DON exposure. Consistently, Que has been reported to ameliorate intestinal barrier disruption by increasing tight junction protein expression in rats or Caco-2 cells [32,33]. To further investigate the role of Que on the epithelial barrier function, we treated IPEC-1 cells with a DON challenge. As expected, DON treatment caused intestinal barrier dysfunction indicated by a reduced TEER and increased FD4 flux from the apical chamber to the basal chamber in IPEC-1 cells. TEER and FD4 permeability are functional parameters for epithelium and tight junctions, which are widely used in assessing barrier integrity in vitro [26]. However, Que treatment improved the TEER and decreased the FD4 permeability from the apical chamber to the basal chamber in IPEC-1 cells after DON challenge. Moreover, Que pretreatment restored the tight junction distribution and abundance after DON treatment. Similarly, our previous study found that Que attenuated the ETEC-K88-induced barrier dysfunction indicated by an increased TEER and decreased FD4 flux in IPEC-1 cells [34]. These data demonstrated that Que protected the intestinal barrier function and integrity after DON exposure in the piglets.

Cell death can lead to intestinal injury [35]. In order to explore the molecular mechanism of the beneficial effect of Que on pigs after DON exposure, we next measured the key signals of the necroptosis signaling pathway. Necroptosis is a special identified pathway of regulated necrosis and is regarded as a highly pro-inflammatory mode of cell death [6]. Various factors can lead to the activation of necroptosis [36]. Initially, the intracellular adapter molecules FADD and TRADD recruit RIP1, which subsequently recruits RIP3 to assemble the necrosome, including phosphorylated RIP1, RIP3, and MLKL. Phosphorylated RIP3 can recruit and phosphorylate the substrate molecule MLKL of RIP3, resulting in programmed cell necrosis [37]. Emerging evidence has demonstrated that necroptosis plays an important role in a variety of physiological and pathological disorders [7,38]. The inhibition of necroptosis minimizes the resulting damage [39]. In the present study, we demonstrated that dietary DON increased the protein expression of t-RIP3 and t-MLKL, which suggests that feeding with DON-contaminated feed activated the necroptosis signaling pathway in the intestine of the piglets. However, supplementation with Que inhibited the increase in t-RIP3 and t-MLKL protein expression. So far, there are few reports about DON and the necroptosis signaling pathway in vivo. Only our previous study found that DON challenge activated the necropoptos signal pathway in IPEC-1 cells [40]. Our experiment firstly demonstrated that DON exposure could activate intestinal cell necroptosis in pigs, and dietary supplementation with Que could inhibit the activation of necroptosis in the intestine of piglets. In order to further investigate the role of Que on necroptosis, we pretreated IPEC-1 cells with Que and then stimulated them with DON challenge. Consistent with in vivo experiment, our results showed that the DON challenge increased the protein abundance of t-RIP1, p-RIP1, t-RIP3, t-MLKL, and p-MLKL in IPEC-1 cells, and Que pretreatment also down-regulated these protein expressions, which demonstrated that Que could suppress the activation of the necroptosis signaling pathway after DON challenge. Until now, the research on Que modulating the necroptosis signaling in the intestine is limited. Most recently, Fan et al. demonstrated that Que prevented necroptosis of oligodendrocytes by inhibiting macrophages/microglia polarization to the M1 phenotype after spinal cord injury in rats [41]. Liu et al. found that Que alleviated the Cd-induced necroptosis in chicken brains through inhibition of the ROS/iNOS/NF-κB pathway [42]. In the present study, it is possible that the protection of Que on intestinal injury and barrier dysfunction might be involved in regulating the necroptosis signaling pathway.

In conclusion, Que exerts protective effects on maintaining the intestinal integrity and barrier function in weaned piglets after DON exposure. These beneficial functions of Que may be associated with decreasing the necroptosis signaling pathway.

## 4. Materials and Methods

### 4.1. Animals and Experimental Design

Animal experiments and procedures were approved by Animal Care and Use Committee of Wuhan Polytechnic University. Twenty-four piglets (Duroc × Large White × Landrace; barrows; initial body weight (BW, 8.91 ± 0.21 kg; 35 ± 1 d of age) were randomly allocated by weight to four treatment groups. Each treatment group was in six replicated pens. Pigs were placed individually in pens (1.80 × 1.10 m). Feed and water were freely available. Pig weight and feed intake were measured on d 1, 7, 14, and 21 for calculation of average daily gain (ADG), average daily feed intake (ADFI), and feed conversion (F/G). Piglets were fed a corn–soybean basal diet. Diets were formulated to meet the nutrient requirements of the NRC (2012).

The in vivo experiment was conducted as a 2 × 2 factorial arrangement, including dietary treatment (basal diet or diet supplemented with 100 mg/kg Que) and DON exposure (basal feed or feed contaminated with 4 mg/kg DON). Que was purchased from Sigma-Aldrich Co., Ltd. (Saint Louis, MO, UAS) and DON was cultivated from *Fusarium graminae* W3008 according to Wang [43]. After feeding with diet supplemented with or without Que for 7 d, DON-contaminated feed or basal feed were put in piglets’ diet for another 21 d. The dose of DON supplemented in feed was selected according to previous studies [14] and our preliminary experiment. The concentration of Que was chosen based on previous studies and our preliminary experiment. At the end of animal experiment, piglets were slaughtered with sodium pentobarbital (80 mg/kg BW) to collect samples.

### 4.2. Cell Culture

The IPEC-1 cell line was derived from mid-jejunum of a neonatal piglet, which was from Texas A&M University. Cells were cultured in Dulbecco’s Modified Eagle’s Medium-F12 (DMEM, HyClone), supplemented with 5% fetal bovine serum (Sigma-Aldrich, St. Louis, MO, USA), 1% insulin-transferrin selenium (Gibco, Carlsbad, CA, USA), 1% penicillin/streptomycin, and epidermal growth factor (5 ng/mL) (Gibco, Carlsbad, California, USA) according to our previous study [40]. Cells were firstly treated with or without 10 μmol/L Que for 24 h, then were stimulated by 0.5 μg/mL DON (Qingdao Puruibang Bioengineering Co., Ltd., Qingdao, China) or PBS for 48 h.

### 4.3. Intestinal Histology

Fixed jejunal samples were prepared using standard paraffin embedding techniques. Three cross sections (4 μm thick) of each intestinal segment were stained with hematoxylin and eosin (H&E). The ten well-oriented, intact villi and their associated crypts from each segment were used to measure the villus height and crypt depth according to our previous study [44].

### 4.4. Intestinal Ultrastructure

Jejunum samples were dehydrated using graded concentrations of ethanol, infiltrated, embedded in Araldite, stained with 4% uranyl acetate and 0.5% lead citrate, and cut into ultrathin sections (100 nm). The ultrastructure of jejunum was observed and imaged using a transmission electron microscope (TEM) (Tecnai, Hillsboro, OR, USA) at an accelerating voltage of 80 kV and a magnification of 8000×.

### 4.5. Cell Viability

IPEC-1 cells were seeded in 24-well plates and firstly pretreated with 0 or 10 μmol /L Que for 24 h, and afterwards treated with or without 0.5 μg/mL DON for another 48 h. Cell viability was measured by CCK-8 kits (Beyotime Institute of Biotechnology, Shanghai, China). To each well was added 10 μL CCK-8 reagent and incubated at 37 °C for 1 h. The absorbance values were obtained at 450 nm by a microplate reader (BioTek Instruments Inc., Winooski, VT, USA) according to the manufacture’s instruction.

### 4.6. Cell Supernatant Lactate Dehydrogenases (LDH)

IPEC-1 cells were seeded in 24-well plates and firstly pretreated with 0 or 10 μmol /L Que for 24 h, and afterwards treated with or without 0.5 μg/mL DON for another 48 h. The cell supernatant was collected to measure LDH activity by using LDH kits (Nanjing Jiancheng, Nanjing, China). LDH activity was evaluated by the absorbance values, which were obtained at 450 nm by a microplate reader (BioTek Instruments Inc., Winooski, USA).

### 4.7. Cell Barrier Function

IPEC-1 cells were grown in collagen-coated permeable polycarbonate filters (Corning, NY, USA). The medium was changed every day until a steady trans-epithelial electrical resistance (TEER) was observed. IPEC-1 cells were seeded in 24-well plates and firstly pretreated with 0 or 10 μmol /L Que for 24 h, and afterwards treated with or without 0.5 μg/mL DON for another 48 h. After treatment, epithelial TEER was measured by Millicell-Electrical Resistance System 2 (Millipore, MA, USA). Data were calculated by subtracting the filter and bathing solution.

After treatments, fluorescein isothiocyanate (FITC)-labeled dextran 4 kDa (FD4, Sigma) was added to the apical compartment at a concentration of 1 mg/mL. Then, monolayers were incubated at 37 °C for 2 h. A total of 50 μL medium from the basolateral compartments were collected and transferred into 96-well black plates (Corning, NY, USA). FD4 fluorescence was measured by a microplate reader (BioTek Instruments Inc., USA) at an excitation wavelength of 490 nm and an emission wavelength of 520 nm. The permeability of monolayer cells was defined as the amount of FD4 that was transported to the basolateral chamber from the apical chamber.

### 4.8. Tight Junction Protein Distribution

Cells were seeded on glass coverslips (Corning, NY, USA) and firstly pretreated with 0 or 10 μmol/L Que for 24 h, and afterwards treated with or without 0.5 μg/mL DON for another 48 h. Cells were fixed with 4% paraformaldehyde and permeabilized with 0.2% Triton X-100. The cells were blocked with 1% bovine serum albumin in PBS for 30 min and then incubated with anti-claudin-1 (#519000, Invitrogen, Waltham, MA, USA) antibody and anti-occludin (#abs131224, Absin) antibody overnight. Cells were then incubated with a goat anti-rabbit secondary antibody conjugated to Alexa Fluor 488 for 2 h (Invitrogen), followed by counterstaining with 4,6-diamidino-2-phenylindole (Sigma-Aldrich, St. Louis, MO, USA). The coverslips were mounted onto glass microscope slides using mounting buffer, and visualized using a fluorescent confocal laser scanning microscope (FV10i, Olympus). The fluorescence intensity was quantified by the Olympus Flouview ver 3.1b software.

### 4.9. Protein Expression by Western Blotting

The total protein from jejunal samples or IPEC-1 cells were homogenized and then measured by a BCA assay kit. Equal amounts of protein samples were separated on sodium dodecyl sulfate-polyacrylamide gels and transferred from the gels onto polyvinylidene fluoride membranes. After blocking with 5% (*w/v*) milk powder in tris-buffered saline containing 0.1% Tween-20 (TBST), the membranes were incubated overnight at 4 °C with appropriate primary antibodies: rabbit anti-Claudin-1 (1:200, Thermo Fisher Scientific, Waltham, MA, USA), rabbit anti-Occludin (1:250, Abcam, Cambridge, UK), rabbit anti-t-RIP1 (1:1000, LifeSpan BioSciences, Seattle, WA, USA), rabbit anti-p-RIP1 (1:2000, Cell Signaling Technology, Boston, Massachusetts, USA), rabbit anti-t-RIP3 (1:1000, Santa Cruz Biotechnology, Santa Cruz, CA, USA), rabbit anti-p-RIP3 (1:2000, Cell Signaling Technology, Boston, MA, USA), mouse anti-t-MLKL (1:1000, Cell Signaling Technology, Boston, MA, USA), rabbit anti-p-MLKL (1:1000, Cell Signaling Technology, Boston, MA, USA), mouse anti-β-actin (1:10,000, Sigma Aldrich, St. Louis, MO, USA). On the following day, the membranes were washed and incubated with secondary antibodies at room temperature. After washing three times, the band intensities were visualized using enhanced Chemiluminescence Western Blot Kit (Amersham, Piscataway, NJ, USA), scanned by Quantity One^®^ software (Bio-Rad Laboratories, CA, USA). The relative abundance of target protein was expressed as the target protein: β-actin ratio. The phosphorylated proteins were also normalized with relative total protein abundance.

### 4.10. Statistical Analysis

Data were analyzed by ANOVA using the general linear model procedures (SAS Institute). The model included the effects of DON, Que, and their interactions. All data were represented as means ± SE. Multiple comparison tests was performed using Duncan’s multiple comparisons after ANOVA analysis. Differences were considered significant for values of *p* ≤ 0.05.

## Figures and Tables

**Figure 1 ijms-24-15172-f001:**
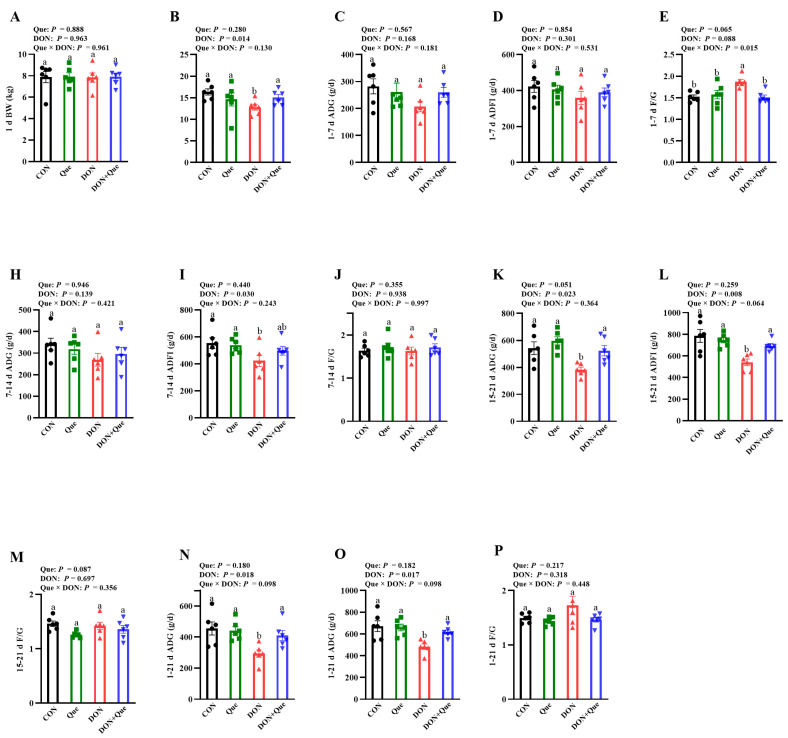
Effects of dietary Que on growth performance of weaned piglets after DON exposure. Weaned piglets were fed either a basal diet or 100 mg/kg Que-supplemented diet and challenged with or without 4 mg/kg DON for 21 days. All the data were analyzed by ANOVA using the general linear model procedures including the effects of DON, Que, and their interactions. Multiple comparison tests were performed using Duncan’s multiple comparisons. Values are means ± SE, n = 6. ^ab^ Different letters represent a significant difference. Differences were considered significant for values of *p* ≤ 0.05. DON: deoxynivalenol; Que: quercetin; BW: body weight; ADG: average daily gain; ADFI: average daily feed intake; F/G: feed: gain ratio.

**Figure 2 ijms-24-15172-f002:**
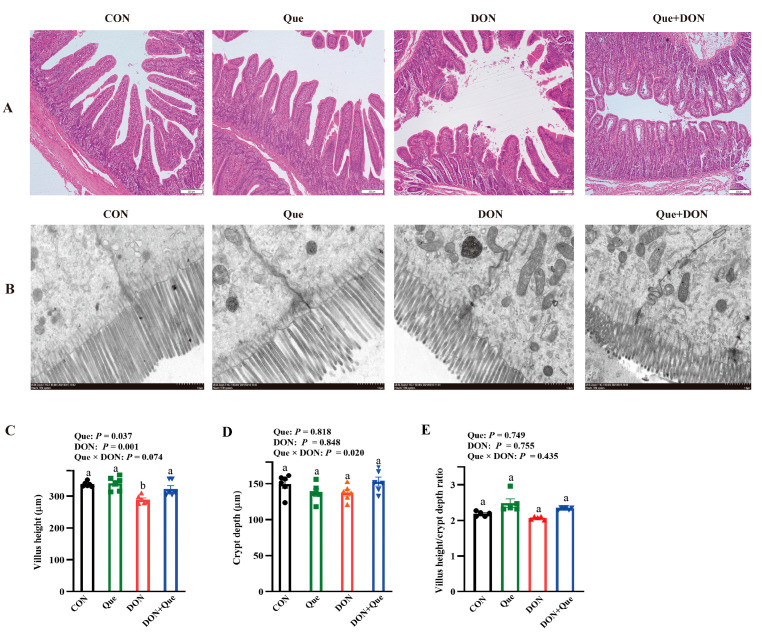
Effects of dietary Que on intestinal morphology in weaned piglets after DON challenge. Weaned piglets were fed either basal diet or 100 mg/kg Que-supplemented diet and then fed with or without 4 mg/kg DON for 21 days. Data were analyzed by ANOVA using the general linear model procedures including the effects of DON, Que, and their interactions. Multiple comparison tests were performed using Duncan’s multiple comparisons. (**A**) Intestinal morphology by HE. (**B**) Intestinal ultrastructure by TEM. (**C**) Villus height. (**D**) Crypt depth. (**E**) Villus height/crypt depth. Values are means ± SE, n = 6. ^ab^ Different letters represent a significant difference. Differences were considered significant for values of *p* ≤ 0.05. HE, hematoxylin-eosin staining; TEM, transmission electron microscope.

**Figure 3 ijms-24-15172-f003:**
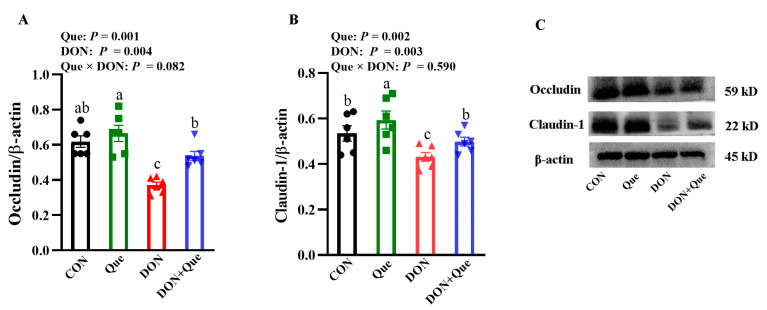
Effects of dietary Que intestinal tight junction protein expression in weaned piglets after DON challenge. Weaned piglets were fed either basal diet or 100 mg/kg Que-supplemented diet and then fed with or without 4 mg/kg DON for 21 days. (**A**,**B**) Tight junction protein expression. (**C**) Representative bands. Values are means ± SE, n = 6. ^a–c^ Different letters represent a significant difference. Differences were considered significant for values of *p* ≤ 0.05.

**Figure 4 ijms-24-15172-f004:**
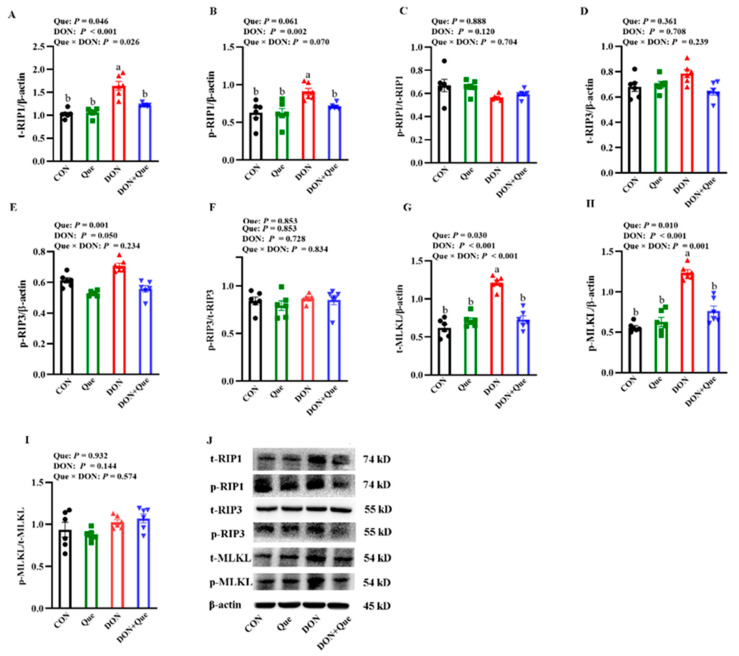
Effects of dietary Que on protein expression of intestinal necroptosis signals in weaned piglets after DON challenge. Weaned piglets were fed either basal diets or 100 mg/kg Que-supplemented diets and then fed with or without DON for 21 days. (**A**–**I**) Protein expression of t-RIP1, t-RIP3, t-MLKL, p-RIP1/t-RIP1, p-RIP3/t-RIP3 and ratio of p-MLKL/t-MLKL. (**J**) Representative bands. Values are means ± SE, n = 6. ^ab^ Different letters represent a significant difference. Differences were considered significant for values of *p* ≤ 0.05. IPEC-1, intestinal porcine epithelial cell 1; t-RIP1, total receptor interacting protein 1; p-RIP1, phosphorylated receptor interacting protein 1; t-RIP3, total receptor interacting protein 3; p-RIP3, phosphorylated receptor interacting protein 3; t-MLKL, total mixed-lineage kinase domain-like protein; p-MLKL, phosphorylated mixed-lineage kinase domain-like protein.

**Figure 5 ijms-24-15172-f005:**
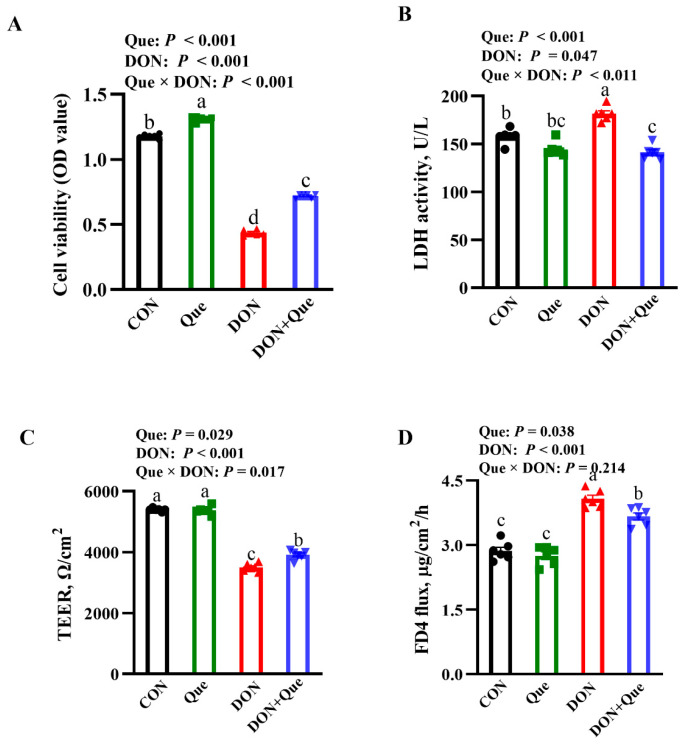
Effects of Que on cell injury and barrier integrity in IPEC-1 cell after DON challenge. Cells were pretreated with 10 µmol/L Que for 24 h and then challenged with or without 0.5 μg/mL DON for 2 h. (**A**) Cell viability. (**B**) LDH activity. (**C**) TEER. (**D**) FD4 permeability Values are means ± SE, n = 6. ^a–d^ Different letters represent a significant difference. Differences were considered significant for values of *p* ≤ 0.05. LDH, lactate dehydrogenases; FD4, fluorescein isothiocyanate (FITC)-labeled dextran 4 kDa; TEER, transepithelial electrical resistance; IPEC-1, intestinal porcine epithelial cell 1.

**Figure 6 ijms-24-15172-f006:**
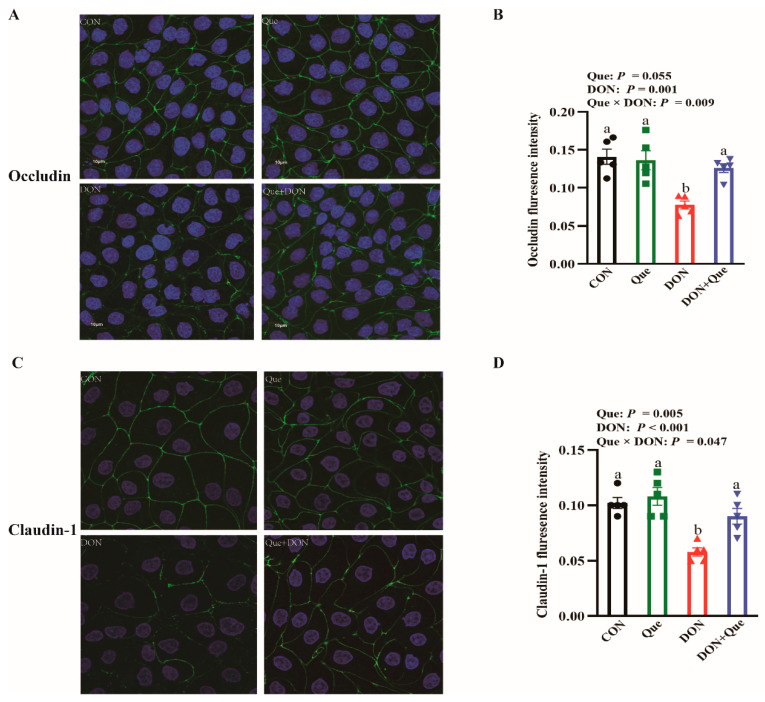
Effects of Que on cellular distribution of tight junction proteins in IPEC-1 cells after DON challenge. IPEC-1 cells were cultured in confocal dish with or without 10 µmol/L Que for 24 h, and then challenged with or without 0.5 μg/mL DON for 2 h. (**A**–**D**) Tight junction protein distribution and fluorescence intensity. The spread and distribution of tight junction proteins were observed through a confocal microscope. Values are means ± SE, n = 6. ^a–c^ Different letters represent a significant difference. Differences were considered significant for values of *p* ≤ 0.05. IPEC-1, intestinal porcine epithelial cell 1.

**Figure 7 ijms-24-15172-f007:**
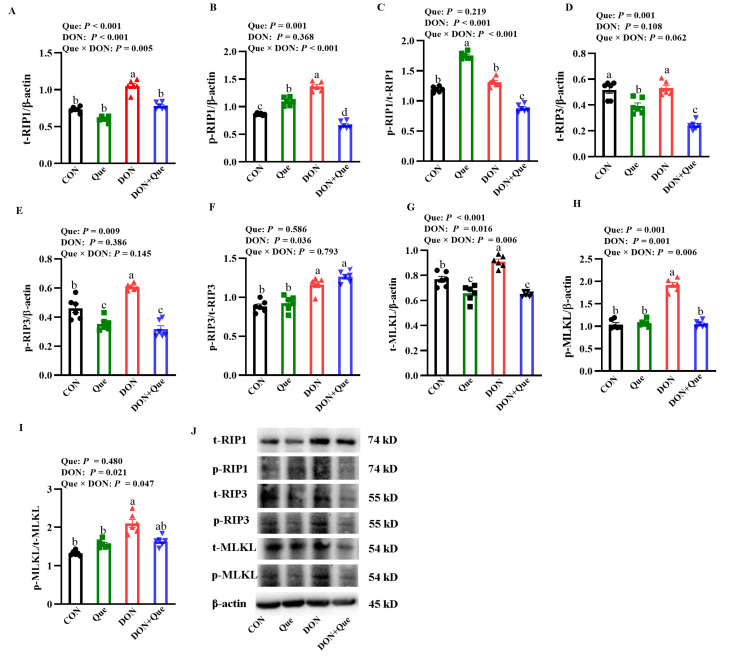
Effects of Que on protein expression of necroptosis signals in IPEC-1 cells after DON challenge. Cells were pre-treated with 10 µmol/L Que for 24 h and then challenged with or without 0.5 μg/mL DON for 2 h. (**A**–**I**) t-RIP1, t-RIP3, t-MLKL, p-RIP1/t-RIP1, p-RIP3/t-RIP3 protein expression, and ratio of p-MLKL/t-MLKL. (**J**) Representative bands. Values are means ± SE, n = 6. ^a–d^ Different letters represent a significant difference. Differences were considered significant for values of *p* ≤ 0.05. IPEC-1, intestinal porcine epithelial cell 1; t-RIP1, total receptor interacting protein 1; p-RIP1, phosphorylated receptor interacting protein 1; t-RIP3, total receptor interacting protein 3; p-RIP3, phosphorylated receptor interacting protein 3; t-MLKL, total mixed-lineage kinase domain-like protein; p-MLKL, phosphorylated mixed-lineage kinase domain-like protein.

## Data Availability

Not applicable.

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
