# Peer review of "Quercetin Ameliorates Deoxynivalenol-Induced Intestinal Injury and Barrier Dysfunction Associated with Inhibiting Necroptosis Signaling Pathway in Weaned Pigs"

_ijms, 2023, doi:10.3390/ijms242015172_

Round 1

Reviewer 1 Report

In this study the authors demonstrate the beneficial effect of Que on intestinal permeability using in vivo and in vitro models. However the topic is exciting and the practical implenetations of findings are realistic, the manuscript could be improved at some points. My comments and suggestions can be read below.

Considered as a whole, the in vitro experiments can be paralleled whith the finding on piglets, the expandation of measurement on intestinal cells or additional models coud strenght the connection between the present findings. For example, the intestinal integrity and barrier function can be easily measured ex vivo, using intestinal sacs filled with non-permeable dyes or by ussing chamber.

Introduction:

- Authors should introduce DON in more detail, describing the source of the toxin, the possible cellular and molecular mechanism of its harmful effect, etc.

- The two consecutive sentences started in line 45 and 47 tell the same information.

- The phrase 'necroptosis is a newly identified form of programmed cell death' has been used very often lately. However, necroptosis is an actively researched topic since the early 2000s. Authors should use an other adjective.

- There are some glitches in the main text, e.g. line 69: " role of    [gap]    Que'.

Results:

- Based on the manuscript template the main text should be in justified format.

- Based on the instructions for authors, the abbreviations should be defined at their first mention. 

- Table and figure legends should contain the applied statistical analysis, particulary while using general terms (e.g. Interaction in table 1.). P values placed over the diagrams should be also defined in figure legends. Several diagram contains abc... letters to indicate the difference between the given groups. I think that those letters should be marked even if there is no significant differences (all group has 'a' letter).

- In some figure legends, in 'abDifferent letters' ab are in supercript. The wrigting style should be consistent.

- Growth performance should be illustrated in graph instead of table data, showing also the time kinetic of BW changes and using more appopriate statistical analysis (paired test). 

- Legend of Figure 2. contains IPEC-1, however, there is no IPEC-1 on that figure. 

- Authors show in Figure 4. that DON treatment resulted in decreased cell viability and increased LDH assay. Based on the description the measurements were performed 2 hours after treatment. I admit that these assays describes the state of the cells at the time of integrity-measurements, but the time kinetic of necroptosis could be wider then 2 hours. An additional experiment, treating cells for 24 hours could be performed to show the final outcome of DON treatment on intestinal cell death. An other comment, that based on the cell viability assay, showing that more than the half of the cells are dead at the time of TEER measurement, the detected barrier disfuncion could be the acute toxic effect of DON instead of altering tight junctions, which was showed in later figures. Authors should have used lower DON concentration, which do not kill the cells so fast. 

- In figure 5., microscopic images show the cells after DON and Que treatment. Based on the previous figure, I would expect that half of the cells are dead. Did the authors experienced any sign of that? Additional staining (e.g. filamental actin with phalloidin) could reveal the early phase of cell death (cytoskeletal disruption).

- Beside occludin and claudin, zonula occludens-1 is also a determinative player in cell adhesion and tight junction. Did the authors try to investigate the intracellular localisation or protein level of ZO-1 in their experiment?

- In the legend of Figure 5, abbreviation of ZO-1 is mentioned, however there is no other appearance of thet molecules nor in the whole manuscript.

- Diagrams in Figure 6. should be consistent (colors, symbols, direction of group titles in x axis, axes fit to a same line, etc.)

Discussion.

- The text needs a thorough review while it contains numerous typos, misspellings, additional or missing punctuation marks or space (e.g. lines 238, 258.)

Author Response

Dear editor and reviewers, Thank you very much for your time and effort to revise our manuscript. We have studied comments carefully and have made correction which we hope meet with approval. Those comments are constructive and very helpful for revising and improving our paper, as well as the important guiding significance to our researches. Revised portion are marked in red in the paper. Here are our responses to your comments. It needs to be mentioned that your comments are marked in blue to distinguish from our responses. Responses to reviewer 1’s Comments 1) In this study the authors demonstrate the beneficial effect of Que on intestinal permeability using in vivo and in vitro models. However, the topic is exciting and the practical implementations of findings are realistic, the manuscript could be improved at some points. My comments and suggestions can be read below. Considered as a whole, the in vitro experiments can be paralleled within the finding on piglets, the explanation of measurement on intestinal cells or additional models could strengthen the connection between the present findings. For example, the intestinal integrity and barrier function can be easily measured ex vivo, using intestinal sacs filled with non-permeable dyes or by using chamber. Response: Thank you for your comments. It is a good ideal that the in vitro experiments paralleled within the finding on piglets, so that the explanation of measurement on intestinal cells could strengthen the connection between the present in vivo findings. We have revised the manuscript according to your advice thorough the revised manuscript. 2)Introduction: authors should introduce DON in more detail, describing the source of the toxin, the possible cellular and molecular mechanism of its harmful effect, etc. Response: Thank you. we have added more information about DON in the introduction in the revised manuscript. For example, “mycotoxins are the most common natural contaminants existing in human food and animal feed. Among the mycotoxins, deoxynivalenol (DON), predominantly generated by Fusarium graminearum and F. culmorum, is often encountered in cereals and cereal products” et al in line 43-45 and line 52-54. 3) The two consecutive sentences started in line 45 and 47 tell the same information. Response: Thank you. We have rewritten this sentence in line 31-34 in the revised MS. 4) The phrase 'necroptosis is a newly identified form of programmed cell death' has been used very often lately. However, necroptosis is an actively researched topic since the early 2000s. Authors should use another adjective. Response: We have change “newly identified” into “specifical” line 56 in the revised MS. 5) There are some glitches in the main text, e.g. line 69: " role of Que Response: Thank you. We have checked all the MS and revised those glitches in the main text such as in line 66,74, 179,243, 247,252,257,318 and 332. 6) Results: Based on the manuscript template the main text should be in justified format. Response: Thank you. We have justified the format of main text based on the manuscript template. 7) Based on the instructions for authors, the abbreviations should be defined at their first mention. Response: Thank you for your kind reminders. We have defined the abbreviations at their first mention in the revised manuscript in the results section, which have marked in line 87-89, line 155-153 and line 155, line 173, line 190-191. 8) Table and figure legends should contain the applied statistical analysis, particulary while using general terms (e.g. Interaction in table 1.). P values placed over the diagrams should be also defined in figure legends. Several diagram contains abc... letters to indicate the difference between the given groups. I think that those letters should be marked even if there is no significant differences (all group has 'a' letter). Response: Thank you. We have added statistical analysis in the table and figure legends and P values over the diagrams has been also defined in the revised figure legends. We have defined the “abc” letters to indicate the difference between the given groups and a letter was marked in the table and figure even if there are no significant differences among the data. 9) In some figure legends, in 'abDifferent letters' ab are in supercript. The wrigting style should be consistent. Response: We are sorry for the mistake. We have made the “ab” letter in superscript in legends of figure 1-5 in the revised MS. 10) Growth performance should be illustrated in graph instead of table data, showing also the time kinetic of BW changes and using more appropriate statistical analysis (paired test). Response: Thank you. We have changed the table 1 into graph in the revised MS. To better describing the results, we used Duncan's multiple comparisons for these data besides the 2×2 factor design by general linear model included DON, Que and their interactions. 11) Legend of Figure 2. contains IPEC-1, however, there is no IPEC-1 on that figure. Response: I am sorry for the mistake. Indeed, there is no IPEC-1 on figure 2 and we have deleted it in the revised manuscript. 12) Authors show in Figure 4. that DON treatment resulted in decreased cell viability and increased LDH assay. Based on the description the measurements were performed 2 hours after treatment. I admit that these assays describe the state of the cells at the time of integrity-measurements, but the time kinetic of necroptosis could be wider than 2 hours. An additional experiment, treating cells for 24 hours could be performed to show the final outcome of DON treatment on intestinal cell death. Another comment, that based on the cell viability assay, showing that more than the half of the cells are dead at the time of TEER measurement, the detected barrier dysfuncion could be the acute toxic effect of DON instead of altering tight junctions, which was showed in later figures. Authors should have used lower DON concentration, which do not kill the cells so fast. Response: In our in vitro experiment, IPEC-1 cells were firstly pretreated with Que for 24 h, and then treated with 0.5 μg/mL DON for another 48 h (in line 389-390). We have added this information in line 383-384 and line 400-402. According to our results, the final outcome of DON treatment was really caused by intestinal cell death. In our preliminary experiment (Supplemental figure 1), different concentration of DON (0, 0.2, 0.5,1 and 2 μg/mL) was used to treated IPEC-1 cells for indicated times (0, 24 and 48 h) and we finally selected the 0.5 μg/mL dose to establish intestinal injury mode. Supplemental figure 1. Effects of different doses of DON on viability of IPEC-1 cells. a-dDifferent letters represent a significant difference. Differences were considered significant for values of P ≤ 0.05. 13) In figure 5., microscopic images show the cells after DON and Que treatment. Based on the previous figure, I would expect that half of the cells are dead. Did the authors experience any sign of that? Additional staining (e.g. filamental actin with phalloidin) could reveal the early phase of cell death (cytoskeletal disruption). Response: After 48 h DON treatment, half of the cells maybe dead by many kinds of cell death. It is a good ideal to stained the filamental actin with phalloidin to reveal the early phase of cell death in our future study. 14) Beside occludin and claudin, zonula occludens-1 is also a determinative player in cell adhesion and tight junction. Did the authors try to investigate the intracellular localisation or protein level of ZO-1 in their experiment? Response: Yes, you are right. Zonula occludens-1 is a determinative player in cell adhesion and tight junction. In fact, we have tried to investigate the intracellular localization or protein level of ZO-1 in our experiment and we failed to gain ideal images because of the low antibody specificity. In future, we will purchase better antibodies to measure this protein. 15) In the legend of Figure 5, abbreviation of ZO-1 is mentioned, however there is no other appearance of thet molecules nor in the whole manuscript. Response: We are sorry for the mistake and we have deleted this word in our MS. 16) Diagrams in Figure 6 should be consistent (colors, symbols, direction of group titles in x axis, axes fit to a same line, etc.) Response: Thank you. we have revised this figure to be consistent including the colors, symbols, direction of group titles in x axis et al. 17) In Discussion. The text needs a thorough review while it contains numerous typos, misspellings, additional or missing punctuation marks or space (e.g. lines 238, 258.) Response: Thank you. We have checked thorough the manuscript and revised the wrong typos, misspellings, additional or missing punctuation marks or space which have marked red in our uploaded revised manuscript. Responses to reviewer 2’s Comments 18) Although authors indicated how they selected the dose of DON based on previous publications and preliminary data, please indicate also how was selected the dose of Que used in vivo in piglets? Response: Yes, you are right. The dose of DON was based on previous publications and preliminary data. Similarly, the dose of Que fed in the feed was chosen according previous studies (1,2) and preliminary experiment. [1] Zou Y I, Wei H K, Xiang Q H, et al. Protective effect of quercetin on pig intestinal integrity after transport stress is associated with regulation oxidative status and inflammation[J]. Journal of Veterinary Medical Science, 2016, 78(9): 1487-1494. [2] Carvalho K M M B, Morais T C, de Melo T S, et al. The natural flavonoid quercetin ameliorates cerulein-induced acute pancreatitis in mice[J]. Biological and Pharmaceutical Bulletin, 2010, 33(9): 1534-1539. 19) Similarly, how were selected the doses of Que and DON used in vitro with IPEC-1 cells? Did the authors perform some dose-dep study of Que with fixes doses of DON and the opposite? Response: As we referred before, the dose of Que and DON was selected by dose-dep study in our preliminary experiment. As showed in supplemental figure 1, different concentration of DON was used to treated IPEC-1 cells for indicated time periods and finally 0.5 μg/mL was chosen to establish the intestinal injury mode. This dose of DON was also published by our previous work such as in Xiao et al 2020. For the Que, we also set different concentration to treat IPEC-1 cells for different time periods (24, 48, 72 h) (Supplemental table 1) and we finally we chose the 10 μM for our further study. This dose of Que was also reported by our recently study published by Xiao et al 2023. [1] Xiao K, Liu C, Qin Q, et al. EPA and DHA attenuate deoxynivalenol‐induced intestinal porcine epithelial cell injury and protect barrier function integrity by inhibiting necroptosis signaling pathway[J]. The FASEB Journal, 2020, 34(2): 2483-2496. [2] Xiao K, Zhou M, Lv Q, et al. Protocatechuic acid and quercetin attenuate ETEC-caused IPEC-1 cell inflammation and injury associated with inhibition of necroptosis and pyroptosis signaling pathways[J]. Journal of Animal Science and Biotechnology, 2023, 14(1): 1-18. Que (μM) 24 h 48 h 72 h 0 1.083 1.179 1.352 2.5 1.068 1.154 1.427 5 1.095 1.158 1.422 10 1.121 1.168 1.469 20 1.132 1.135 1.453 40 0.994 1.066 1.189 80 0.792 0.755 0.628 100 0.773 0.715 0.555 Supplemental table 1 Effects of different doses of Que on viability of IPEC-1 cells.

Reviewer 2 Report

Dear Editor, Dear Authors,

I was invited to evaluate the manuscript « Quercetin ameliorates deoxynivalenol-induced intestinal injury and barrier dysfunction associated with inhibiting necroptosis signaling pathway in weaned pigs » by Jiahao Liu et al.

In this study, the authors report on the protective effect of quercetin, a food-associated molecule, against toxicity of deoxynivalenol, also found in food, in term of necroptosis in vivo in piglets. For that, the authors used twenty-four weaned piglets that were exposed to Que (basal diet or diet supplemented with 100 mg/kg Que) and DON exposure (control feed or feed contaminated with 4 mg/kg DON). In parallel, they used the in vitro model IPEC-1 that was pretreated with or without Que (10 μmol/mL) in the presence or absence of DON challenge (0.5 μg/mL). In vivo data indicate a protective effect of Que on DON-induced injuries and affects the modulation of various factors by DON : t-RIP1), phosphorylated RIP1 (p-RIP1), p-RIP3, total mixed lineage kinase domain-like protein (t-MLKL) and p-MLKL. Similar results were obtained in vitro on IPEC-1 with Que pretreatment increasing cell viability, decreasing lactate dehydrogenases (LDH) released and barrier function dysfonction caused by DON. In accordance with in vivo data, in IPEC-1 pretreatment with Que also inhibited the protein abundance of t-RIP1, p-RIP1, t-RIP3, p-RIP3, t-MLKL and p-MLKL after DON challenge. Conclusions from the authors are that Que can ameliorate DON-induced intestinal injury and barrier dysfunction associated with suppressing necroptosis signaling pathway at least in pig and in vitro model of pig intestine.

Overall, I found the study well designed and well conducted.

Please find below few minor comments :

1- Although authors indicated how they selected the dose of DON based on previous publications and preliminary data, please indicate also how was selected the dose of Que used in vivo in piglets ?

2- Similarly, how were selected the doses of Que and DON used in vitro with IPEC-1 cells ? Did the authors performed some dose-dep study of Que with fixes doses of DON and the opposite ?

Regards

Author Response

Dear reviewers, Thank you very much for your time and effort to revise our manuscript. We have studied comments carefully and have made correction which we hope meet with approval. Those comments are constructive and very helpful for revising and improving our paper, as well as the important guiding significance to our researches. Revised portion are marked in red in the paper. Here are our responses to your comments. Responses to reviewer 2’s Comments 1) Although authors indicated how they selected the dose of DON based on previous publications and preliminary data, please indicate also how was selected the dose of Que used in vivo in piglets? Response: Yes, you are right. The dose of DON was based on previous publications and preliminary data. Similarly, the dose of Que fed in the feed was chosen according previous studies (1,2) and preliminary experiment. [1] Zou Y I, Wei H K, Xiang Q H, et al. Protective effect of quercetin on pig intestinal integrity after transport stress is associated with regulation oxidative status and inflammation[J]. Journal of Veterinary Medical Science, 2016, 78(9): 1487-1494. [2] Carvalho K M M B, Morais T C, de Melo T S, et al. The natural flavonoid quercetin ameliorates cerulein-induced acute pancreatitis in mice[J]. Biological and Pharmaceutical Bulletin, 2010, 33(9): 1534-1539. 2) Similarly, how were selected the doses of Que and DON used in vitro with IPEC-1 cells? Did the authors perform some dose-dep study of Que with fixes doses of DON and the opposite? Response: As we referred before, the dose of Que and DON was selected by dose-dep study in our preliminary experiment. As showed in supplemental figure 1, different concentration of DON was used to treated IPEC-1 cells for indicated time periods and finally 0.5 μg/mL was chosen to establish the intestinal injury mode. This dose of DON was also published by our previous work such as in Xiao et al 2020. For the Que, we also set different concentration to treat IPEC-1 cells for different time periods (24, 48, 72 h) (Supplemental table 1) and we finally we chose the 10 μM for our further study. This dose of Que was also reported by our recently study published by Xiao et al 2023. [1] Xiao K, Liu C, Qin Q, et al. EPA and DHA attenuate deoxynivalenol‐induced intestinal porcine epithelial cell injury and protect barrier function integrity by inhibiting necroptosis signaling pathway[J]. The FASEB Journal, 2020, 34(2): 2483-2496. [2] Xiao K, Zhou M, Lv Q, et al. Protocatechuic acid and quercetin attenuate ETEC-caused IPEC-1 cell inflammation and injury associated with inhibition of necroptosis and pyroptosis signaling pathways[J]. Journal of Animal Science and Biotechnology, 2023, 14(1): 1-18. Dose of Que (μM) 24 h 48 h 72 h 0 1.083 1.179 1.352 2.5 1.068 1.154 1.427 5 1.095 1.158 1.422 10 1.121 1.168 1.469 20 1.132 1.135 1.453 40 0.994 1.066 1.189 80 0.792 0.755 0.628 100 0.773 0.715 0.555 Supplemental table 1 Effects of different doses of Que on viability of IPEC-1 cells.

Round 2

Reviewer 1 Report

Thank you for the detailed answers.